# DeepLayout: Learning Neural Representations of Circuit Placement Layout

Yuxiang Zhao [1 2]  Zhuomin Chai [1 3]  Xun Jiang [1]  Qiang Xu [2 4]  Runsheng Wang [1 5 6]  Yibo Lin [1 5 6]

## Abstract

Recent advancements have integrated various deep-learning methodologies into physical design, aiming for workflows acceleration and surpasses human-devised solutions. However, prior research has primarily concentrated on developing task-specific networks, which necessitate a significant investment of time to construct large, specialized datasets, and the unintended isolation of models across different tasks. In this paper, we introduce DeepLayout, the first general representation learning framework specifically designed for backend circuit design. To address the distinct characteristics of post-placement circuits, including topological connectivity and geometric distribution, we propose a hybrid encoding architecture that integrates GNN with spatial transformers. Additionally, the framework includes a flexible decoder module that accommodates a variety of task types, supporting multiple hierarchical outputs such as nets and layouts. To mitigate the high annotation costs associated with layout data, we introduce a mask-based self-supervised learning approach designed explicitly for layout representation. This strategy involves a carefully devised masking approach tailored to layout features, precise reconstruction guidance, and most critically—two key supervised learning tasks. We conduct extensive experiments on large-scale industrial datasets, demonstrating that DeepLayout surpasses state-of-the-art (SOTA) methods specialized for individual tasks on two crucial layout quality assessment benchmarks. The experiment results underscore the framework's robust capability to learn the intrinsic properties of circuits.

[1]Peking University [2]National Technology Innovation Center for EDA [3]Wuhan University [4]The Chinese University of Hong Kong [5]Institute of Electronic Design Automation, Wuxi, China [6]Beijing Advanced Innovation Center for Integrated Circuits . Correspondence to: Yibo Lin <yibolin@pku.edu.cn>.

*Proceedings of the 42nd International Conference on Machine Learning*, Vancouver, Canada. PMLR 267, 2025. Copyright 2025 by the author(s).

## 1. Introduction

Modern chip design has become extremely complex due to the continuous advancement of manufacturing technology and the ever-increasing scale of circuits (e.g. TPU, GPU). Recently, advanced AI methods have been widely employed across various stages of circuit design, significantly enhancing both automation and intelligence (DSO.ai, 2020; JedAI, 2022). In the back-end (physical design) domain, researchers have successfully leveraged AI for tasks such as macro placement (Mirhoseini et al., 2020; Chen et al., 2023; Lai et al., 2023; 2022), parameter tuning (Luo et al., 2024; Hsiao et al., 2024; Geng et al., 2023), and layout hotspot detection(Shao et al., 2024; Geng et al., 2022), thereby accelerating design cycles and, in certain instances, achieving performance that surpasses human-devised solutions.

Fig. 1 shows chip design flow. To reduce reliance on time-intensive EDA tools, the aforementioned methods construct task-specific datasets and train corresponding circuit representation models to enable efficient assessment of circuit performance. Constructing such large-scale, specialized, and labeled datasets requires extensive domain expertise and significant human resources. However, some raw circuit designs can be acquired from these sources: (1) open-

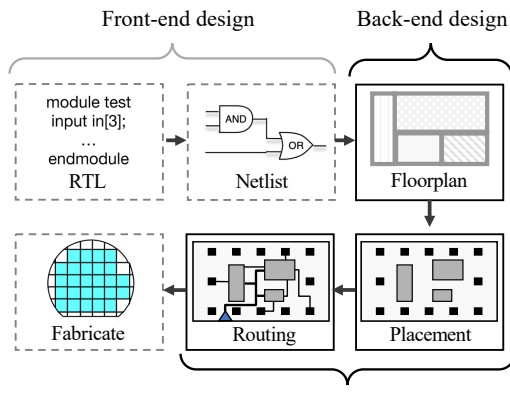

*Figure 1.* The chip design flow is primarily divided into front-end functional/logic design and back-end physical design. Compared to front-end design, the back-end phase is significantly more time-consuming. The finalized chip layout is sent to the foundry for fabrication.

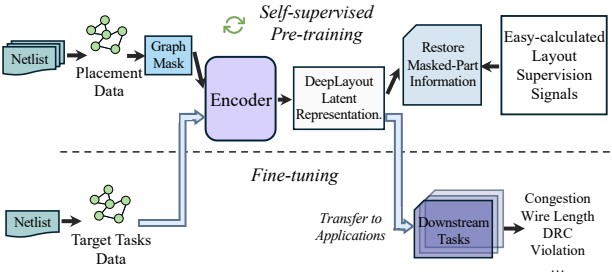

*Figure 2.* Illustration of the DeepLayout paradigm for circuit layout representation. This paradigm enables self-supervised pre-training on placement-only datasets and applies the learned representations to multiple applications.

source repositories (e.g., GitHub), (2) intermediate design parameter configurations, and (3) legacy design databases. These unlabeled circuit data can potentially be leveraged using self-supervised learning techniques—demonstrated to be highly effective in the image (He et al., 2022b) and text (Devlin et al., 2018)—to extract strong and general representations. Motivated by these considerations, we pose a key question: *Can we harness massive collections of raw circuit data to deeply uncover intrinsic circuit properties thereby yielding a general model that generalizes across a broad spectrum of downstream tasks?*

In recent years, some research has focused on applying circuit representation learning to front-end (logic) design by transforming circuits into AIG (And-Inverter Graph) structures and leveraging unsupervised or contrastive learning strategies to uncover latent relationships between circuit functionality and logic (Li et al., 2022; Shi et al., 2023; 2024; Wang et al., 2022b). However, these methods exhibit significant limitations when extended to back-end circuit representation tasks. As circuit logic functionality is not strongly correlated with circuit performance, existing approaches often fail to adequately emphasize the extraction of physical attributes relevant to PPA (Power, Performance, Area), thus limiting their applicability to back-end design scenarios.

To address these challenges, we propose a layout-oriented masked autoencoder, called **DeepLayout**, for back-end self-supervised representation learning. Fig. 2 illustrate the DeepLayout framework. We model the post-placement circuit as a heterogeneous graph enriched with geometric information, employing node and net as two distinct node types to capture the circuit's hierarchical structure. Compared with masking strategies for images or generic graphs, a layout-oriented masked autoencoder must satisfy three key requirements. ***Firstly***, both geometric and topological information are vital in post-placement circuits. For example, nodes that are topologically adjacent may be widely separated in physical space. Therefore, we develop a refined

network that progressively captures geometric and topological relationships, providing the rich feature representations essential for downstream tasks at both node and layout levels. ***Secondly***, unlike front-end representation emphasizing functional attributes, back-end design primarily targets geometric characteristics rather than functional behavior. As a result, simply masking irrelevant nodes at random and restoring node-category attributes is inadequate. We adopt a size-adaptive masking approach based on grid units to ensure that masked grid regions retain consistent macroscopic attributes, such as cell density. ***Thirdly***, to foster synergy between local and global feature reconstruction, we go beyond restoring the masked nodes' coordinates by incorporating additional routing-related features to enhance the network's representational capacity. Specifically, we introduce an easy-calculated intermediate routing feature (RPA), enabling the encoder to capture and exploit routing information and thus mitigate potential biases caused by particular routing algorithms.

Our contributions are as follows:

- We are the first to propose a general circuit representation framework for placement layout learning, enabling fine-grained feature learning from unlabeled data through novel mask strategies and self-supervised tasks.

- We propose a hybrid encoder that jointly models topological connectivity and geometric layouts via heterogeneous GNNs and spatial attention mechanisms. The pre-trained encoder demonstrates remarkable transferability across diverse downstream tasks.

- We conduct extensive experiments on industrial-scale benchmarks, demonstrating DeepLayout's superiority over SOTA methods in congestion prediction and post-routing wirelength estimation.

## 2. Preliminary

### 2.1. Masked-based Representation Pre-training

There has been a longstanding desire to achieve a deeper data understanding without relying on human-annotated labels. Masked-based representation pre-training, a self-supervised learning paradigm, addresses this challenge by employing a process of *Masking* and *Restoration*.

In this approach, certain portions of the data are masked, and the model is trained to restore the missing parts based on the visible ones. By learning to predict these hidden components, the model can develop more meaningful and generalizable representations, enhancing its ability to perform well on downstream tasks. This concept underpins many state-of-the-art models across a variety of domains,

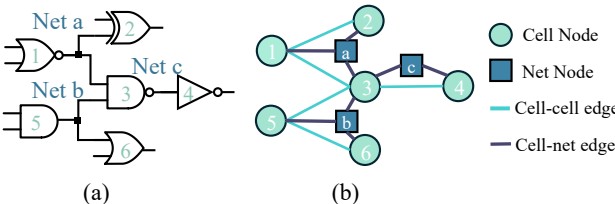

(a)     (b)

*Figure 3.* Illustration of circuit heterogeneous graph. (a) Original circuit netlist with cells and connecting nets. (b) Corresponding heterogeneous graph, where cells and nets are represented as distinct node types, connected by cell-cell and cell-net edges to capture both structural and hierarchical information.

such as BERT (Devlin et al., 2018), where masked words or tokens are predicted, and Masked Autoencoders (MAE) (He et al., 2022b), which mask image patches and train the model to restore them. The pre-trained models can thereby be fine-tuned for specific tasks, such as classification, segmentation, or object detection, yielding superior results. This masked-based pre-training paradigm is particularly well-suited for scenarios where obtaining negative samples is challenging, such as in circuit representation. It enables the model to approximate the underlying data distribution and learn intrinsic data features without the need for explicit labels, thus making it a powerful tool for enhancing representation learning in these contexts.

## 2.2. Circuit representation learning

Circuit representation learning has recently emerged as a pivotal research direction for enabling machine learning-driven analysis and optimization of electronic circuits (Chen et al., 2024). In the realm of front-end design, the representation of circuits commonly involves the utilization of the synthesized netlist to depict the internal configuration of the circuit. The concept introduced in a study by Wang et al. (2022b) aimed to investigate netlist representation, proposing a contrastive learning framework that allows the network to comprehend both the structural elements and embedding functionality of netlists. Subsequently, Deep-Gate (Li et al., 2022) approached this issue from a different angle by developing neural representations of logic gates, incorporating signal probability supervision to account for both the structural characteristics and logical operations of circuits for various downstream applications. Building upon this work, a subsequent study by Shi et al. (2023) sought to overcome the limitations of its precursor by implementing a more direct supervision strategy utilizing pair-wise differences in truth tables, thereby enhancing the ability to capture circuit functionality more effectively. However, as the logical functionality of a circuit is not closely tied to performance, these function-oriented front-end representation approaches are not applicable in back-end circuits.

## 3. DeepLayout

Achieving effective pre-training requires careful consideration of two critical factors: the learning strategy and the supervision signal. For now, few studies attempt to apply mask-style self-supervised representation learning methods to circuits after placement. We attribute this to two primary challenges: (i) the difficulty in selecting suitable objects to mask (e.g., nodes? modules?), and (ii) it is challenging to simply use the original input information as the restoration target that is commonly done in vision tasks.

Considering this, our DeepLayout introduces a novel framework that paves the way for the achievement of *mask-based* self-supervised circuit representation pre-training. The organization is as follows: Section 3.1 introduces a graph data structure used to represent post-placement circuits. Section 3.2 and Section 3.3 discuss the masking strategy and the two pre-trained supervision tasks. Section 3.4 presents the network architecture design.

### 3.1. Circuit Heterogeneous Graph

**Graph Structure.** We start by modeling the circuit netlist as a graph. Fig. 3(a) represents the circuit's logical connectivity through cells (i.e., AND/OR gates or flip-flops) and nets (connecting cell terminals). We represent each cell as a node $v \in V$ in the graph, while edges encode predefined physical or interconnect relationships between connected cells. To address the inherent hierarchy of circuit blocks (e.g., arithmetic units), we introduce virtual net nodes $u \in U$ that aggregate connectivity patterns within sub-circuits. This dual-node formulation transforms into a heterogeneous graph (Fig. 3(b)) that explicitly preserves both structural connectivity and functional hierarchy. To this end, we maintain three critical circuit properties— including signal propagation paths, fanout/fanin relationships, and block-level abstraction.

**Graph Feature.** We formally define the heterogeneous graph $G = \{V, U, E, X_v, X_u\}$, where $E$ represent edges that encode cell-to-cell direct dependencies and cell-to-net membership. The feature matrices $X_v$ and $X_u$ contain cell and net features. $X_v$ including cell size and spatial coordinates, which jointly govern routing related PPA learning. $X_u$ storing the connected pins degree and net span on each axis. Here, $n_v$ and $n_u$ denote the total cells and nets in the design. Tab. 1 shows the graph features in detail. Notably, we omit edge features to avoid over-engineering connectivity patterns. Unlike prior works (Yang et al., 2022; Wang et al., 2022a) that relied on handcrafted features like local cell density or pin density, We pose DeepLayout operates directly on the circuit's native structural and geometric attributes. This benefits DeepLayout by eliminating tedious task-specific preprocessing, enabling a unified circuit representation adaptable to diverse physical design downstream

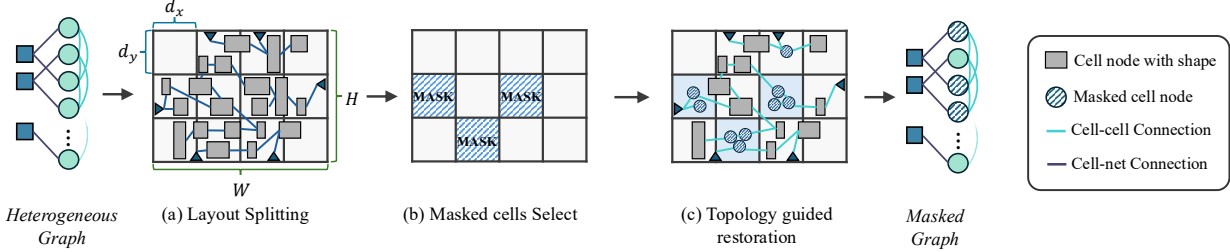

*Heterogeneous Graph*  (a) Layout Splitting  (b) Masked cells Select  (c) Topology guided restoration  *Masked Graph*

*Figure 4.* The pipeline of layout-oriented masking strategy. Cell-to-net heterogeneous relation also retained but are hidden in figure for simplify.

*Table 1.* The features utilized in circuit heterogeneous graph.

| Node | Type | Feature | Definition |
|------|------|---------|------------|
| | Cell | $h$ | Cell Height |
| | | $w$ | Cell Width |
| $v$ (cell) | | $p_n$ | Number of pins |
| | Geometric | $x$ | Cell coordinate in X asix |
| | | $y$ | Cell coordinate in Y asix |
| $u$ (net) | Connections | $d$ | Net degree |
| | Span | $h_u$ | Net span in Y asix |
| | | $w_u$ | Net span in X asix |

tasks.

### 3.2. Layout-oriented Masking

Fig. 4 depicts the masking strategy in DeepLayout. We systematically demonstrate this through modular decomposition, as outlined below.

**Layout Splitting** is inspired by He et al. (2022b). Since circuit layouts can be represented as images, we are in line with this philosophy by partitioning the layout into a set number of non-overlapping grids $\{g\}$, treating all cells within the same grid as an independent masked entity (Fig. 4(a)). Specifically, we pre-define the single grid size as $d_x \times d_y$ and the layout plane, with dimensions $W \times H$, is split into $\frac{W}{d_x} \times \frac{H}{d_y}$ grids, denoted as $\{g_{i,j}\}$. Each cell $c_k(c_k = v, v \in V)$ is then aligned to a corresponding grid $g_{i,j}$ based on its coordinates $(x_k, y_k)$.

$$g_{i,j} = \left\{ c_k \mid \lfloor x_k/d_x \rfloor = i, \ \lfloor y_k/d_y \rfloor = j \right\}, \quad (1)$$

where $i, j$ represent the grid indices ($1 \leq i \leq \frac{W}{d_x}, 1 \leq j \leq \frac{H}{d_y}$), and $\lfloor x \rfloor$ symbolizes the floor function of coordinate $x$. Unlike the fixed patch numbers in image-based models like ViT (Dosovitskiy et al., 2020), DeepLayout employs a variable number of grids for each design. This offers two distinct advantages: (i) it accommodates a wide range of floorplan sizes, from small modules to full-scale SoCs, and (ii) it guarantees a relatively uniform distribution of cells

---

**Algorithm 1** Layout-oriented Masking Algorithm.

**Input:** Heterogeneous graph $G$, input layout size $W \times H$, pre-defined mask ratio $\gamma$, grid size $dx \times dy$.

**Output:** Masked graph and corresponding processed features.

1  Extract the coordinates $(x, y)$ of each cell node in the graph
2  Partition the layout of size $W \times H$ into grid set $\{g\}$ using fixed grid size $dx, dy$
3  Align each cell to the corresponding grid according to its coordinates $(x, y)$;
4  Count the number of non-empty grids $N$ and compute the number of grids to be masked $N_m = N \times \gamma$
5  Randomly select the grids to be masked: $\{g_m\} \leftarrow$ RANDOM $(\{g\}, N_m)$;
6  Initialize the set of masked cell node indices: $I_m = \varnothing$
7  **for** $i = 1 \rightarrow N_m$ **do**
8     **for** *each cell $\in$ the current grid $g_i$* **do**
9        Add the cell index to $I_m$: $I_m \leftarrow I_m \cup \{\text{index}\}$
10  Extract the cell node $c_k$ features $X_v$ of the graph $G$, $c_k = v, v \in V$
11  Copy cell features $X_v^{\text{processed}} \leftarrow X_v$
12  **for** *each index $v_i \in I_m$* **do**
13     Padding cell node coordinates $x, y$ to 0 Set $X_{v_i}^{\text{processed}} \leftarrow [0, 0, h, w, p]$
14  Replace the original features of the graph $G$ with the processed features $X_v^{\text{processed}}$

---

within each grid, maintaining consistent physical characteristics from a global layout perspective. We empirically set the grid size $d_x, d_y$ to correspond to 30 routing track lengths. In some corner cases where standard cells span across multiple grid boundaries, we assign them to the grid where they have the largest coverage area.

**Masked Grid Selection** Unlike image masking methods that permit arbitrary patch selection across the entire canvas, the post-layout floorplan includes vacant regions devoid of cells, necessitating the selection of masked grids exclusively from non-empty candidates ($g_{i,j} \neq \varnothing$). We designate the selected candidates as masked grids $\{g_{i,j}^m\}$, with the

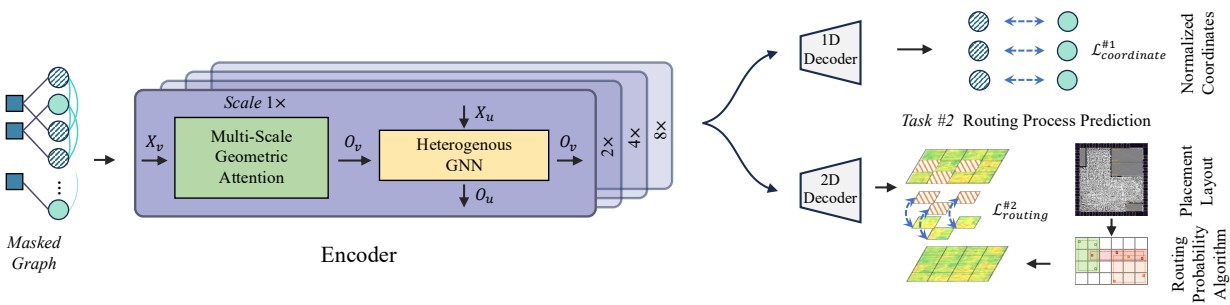

*Figure 5.* Overview of the DeepLayout pre-training pipeline. DeepLayout encodes a masked heterogeneous graph using multi-scale geometric attention and heterogeneous GNNs at increasing spatial resolutions $[1\times, 2\times, 4\times, 8\times]$. It is trained via two objectives: cell coordinate reconstruction and routing process prediction, optimized by $\mathcal{L}^{\#1}_{coordinate}$ and $\mathcal{L}^{\#2}_{routing}$.

remainder classified as visible grids $\{g^v_{i,j}\}$. We hold the hyperparameter $\gamma$ to balance between these two subsets. Thus, visible and masked cells are aggregated by combining cells from $\{g^m_{i,j}\}$ and $\{g^v_{i,j}\}$, respectively, formalized as: $\{c^v_k\} = \bigcup_{i,j} g^v_{i,j}$ and $\{c^m_k\} = \bigcup_{i,j} g^m_{i,j}$. The graph $G$ is subsequently updated by replacing cell node features with processed features $\{X^{processed}_{c_k}\}$. For cells in masked grids $\{c^m_k\}$, the coordinate values $(x_k, y_k)$ are padded to 0 while retaining the original size and property features $[h, w, p]$.

$$
X^{processed}_{v_k} = \begin{cases} [0, 0, h, w, p] & \text{if } c_k \in \{c^m_k\}, \\ X_{v_k} & \text{if } c_k \notin \{c^m_k\}. \end{cases} \quad (2)
$$

Mask ratio $\gamma$ is typically determined empirically. We can cast the previous unsupervised pre-training works (Zhu et al., 2022; Li et al., 2022) as a special case of DeepLayout, where the $\gamma$ is fixed at 0 to enable autoregressive learning. In practice, we found that setting $\gamma$ to 50% yields optimal pre-training performance, which contrasts with the higher 75% mask ratio commonly used in MAE. We hypothesize that this difference arises due to the nature of the data. Images tend to exhibit a high level of information redundancy. In contrast, circuits contain rich information among subcircuits.

**Restoration Guidance** is another crucial aspect to consider. Typically, circuits exhibit weaker global correlations than images. This makes it particularly challenging for the network to accurately reconstruct masked regions from the visible parts—akin to attempting to restore a QR code without any prior knowledge. To address this, we preserve the original topological connections between masked cells $(v \to v)$ and cells-to-nets $(v \to u)$ to aid in the restoration process. Fig. 4(c) illustrates masked graph with topological connections. From a physical design standpoint, this approach is analogous to the incremental placement or Engineering Change Order (ECO) process, where local modifications are made while maintaining the overall circuit design integrity.

### 3.3. Pre-training Supervision Signal

The supervision signal plays a critical role in determining the knowledge acquired during pre-training. Fig. 5 illustrates the pre-training framework of DeepLayout, which incorporates two carefully designed tasks to exploit the unique properties of post-layout circuits: #1 Cell coordinate reconstruction: recover masked cell coordinates. #2 Routing process prediction: modeling interconnect behavior.

**Task #1 Cell Coordinate Reconstruction.** Placed cell coordinates provide essential geometric constraints for physical design's downstream tasks. We implement a coordinate reconstruction objective through a single-layer multilayer perceptron (MLP) that predicts normalized coordinates of masked grid nodes. Formally, for each masked node $\{c^m_k\}$, the MLP generates predicted coordinates $\hat{P} = \{\hat{p}_k | k = 1, 2, ..., K\}$, while the ground-truth normalized coordinates are denoted as $P = \{p_k | k = 1, 2, ..., K\}$. The reconstruction loss is computed using the mean squared error (MSE) between $\hat{P}$ and $P$ as formulated below:

$$
\mathcal{L}^{\#1}_{coordinate} = \frac{1}{K} \sum_{p_k \in P} \| p_k - \hat{p}_k \|^2_2 \quad (3)
$$

**Task #2 Routing Process Prediction.** Compared to cross-stage evaluation from the placement, routing is time-intensive yet provides more accurate performance metrics, especially for timing and power consumption. However, directly training a machine learning model to mimic the routing process is highly challenging. To address this, we guide the model to learn an intermediate routing representation that captures essential aspects of the routing process. Specifically, we use a computationally efficient Routing Probability Algorithm (RPA) as the supervision signal. This algorithm estimates the probability of routing paths within the net bounding box of a sub-circuit, defined as follows:

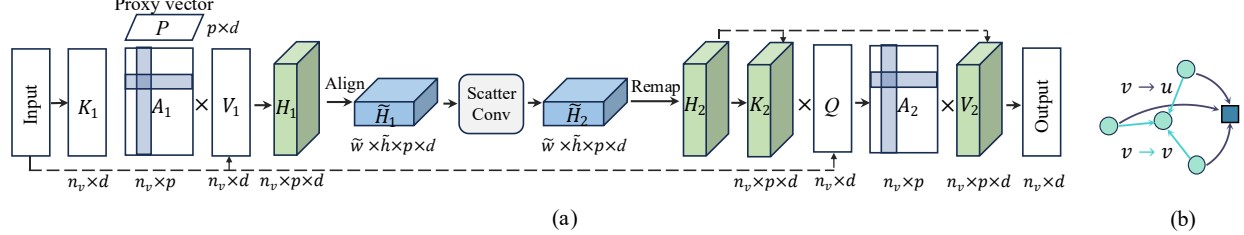

*Figure 6.* Encoder's geometric and topological module: (a) Multi-scale Geometric Attention (MSGA), (b) Heterogeneous GNN (HGNN).

$$\text{RPA}_u(x,y) = \left(\frac{1}{x_u^h - x_u^l} + \frac{1}{y_u^h - y_u^l}\right). \quad (4)$$

where $x_u, y_u$ represents the cell or pin positions within the net u, and $x_u^h = \max x_u, x_u^l = \min x_u, y_u^h = \max y_u, y_u^l = \min y_u$ denotes the net bounding box. And $\text{RPA}_u(x,y)$ outside the region $[x_u^l, x_u^h, y_u^l, y_u^h]$ set to 0. The total routing probability across the layout is then computed as:

$$\text{RPA} = \sum_{0<x<W, 0<y<H, u \in U} \text{RPA}_u(x,y). \quad (5)$$

where $W, H$ is corresponds to the layout size and $RPA \in \mathbb{R}^{W \times H}$. This RPA is similar to the algorithm introduced in Spindler & Johannes (2007). The objective is defined as minimizing the MSE between the $R\hat{P}A$ among masked nodes and the ground-truth $RPA$, formulated as:

$$\mathcal{L}_{\text{routing}}^{\#2} = \frac{1}{N_m} \sum_{i=1}^{N_m} \left\| \mathbf{RPA}_i - \mathbf{R\hat{P}A_i} \right\|_2^2 \quad (6)$$

where $N_m$ denotes the number of masked grids. We summarize the benefits of using RPA in twofold: (i) it allows routing related features to be derived solely from placement information, without the need for additional labels, thereby simplifying computation; and (ii) it helps the model generalize better and reduces bias caused by the label randomness inherent in heuristic routing algorithms.

**Overall Loss.** To this end, the overall loss function for DeepLayout's self-supervised pre-training is formulated by integrating the two component losses:

$$\mathcal{L} = \mathcal{L}_{coordinate}^{\#1} + \mathcal{L}_{\text{routing}}^{\#2} \quad (7)$$

Throughout the pre-training phase, we iteratively minimize the loss until the model attains numerical convergence.

### 3.4. Layout Representation Network

We instantiate DeepLayout network, with an asymmetric architecture consisting of a task-agnostic encoder and task-specific decoders. This flexible architecture ensures unified pre-training during the physical design stage and facilitates

application across multiple downstream tasks. We introduce each component as follows:

**Encoder.** The encoder is designed as cascaded architecture, as shown in Fig. 5(a), which alternately stacked geometric and topological modules. It operates on all graph's nodes and encodes each node's feature into the latent space. During both pre-training and fine-tuning, the encoder's architecture remains unchanged. A detailed description of each module is in the following parts.

- *Multi-scale Geometric Attention (MSGA)* captures spatial relationships between standard cells in geometric space by combining local and global perspectives, as illustrated in Fig. 5(b). In the local perspective, we aggregate multi-scale sparse convolution features from 2D-mapped cells. In the global perspective, to address the computational complexity $\mathcal{O}(n^2 d)$ of native transformer attention mechanisms (where $n$ is often large in circuits), MSGA reduces complexity by decomposing global attention into two independent parts, inspired by He et al. (2022a). The MSGA module operates exclusively on the standard cell features $X_v \in \mathbb{R}^{n_v \times d}$ of graph $G$, where $d$ is the feature dimension.

The first part of global attention is generated by applying a Multi-Layer Perceptron (MLP) to linearly project $X_v$, producing key ($K_1$) and value ($V_1$) vectors. To address the inefficiency of directly using the native projected query from $X_v$ in attention calculations, we introduce a learnable proxy query $P \in \mathbb{R}^{p \times d}$, where $p$ (the length of $P$) is much smaller than $n_v$. The first attention matrix $A_1$ is computed using $P$ and $K_1$, and the hidden feature $H_1$ is obtained by multiplying $V_1$ with $A_1$, $K_1 \in \mathbb{R}^{n_v \times d}, V_1 \in \mathbb{R}^{n_v \times d}$. The formulation for the first attention is as follows:

$$K_1, V_1 = \text{MLP}_K(X_v), \text{MLP}_V(X_v), \quad (8a)$$

$$A_1 = \text{Softmax}(K_1 P^T), \quad A_1 \in \mathbb{R}^{n_v \times p}, \quad (8b)$$

$$H_1 = A_1^T V_1, \quad H_1 \in \mathbb{R}^{n_v \times p \times d}, \quad (8c)$$

Next, we aggregate multi-scale local features by mapping the hidden feature $H_1$ to a 2D plane using the

scale parameter $s$, as outlined in line 3 of Section 3.2. This produces the scale-mapped feature $\tilde{H}_1$, where $s$ aligns with the grid size $dx \times dy$ from the layout partitioning algorithm. The feature dimensions $\tilde{w} \times \tilde{h}$ are calculated as $\tilde{w} = \frac{W}{s \times dx}$ and $\tilde{h} = \frac{H}{s \times dy}$, with $W$ and $H$ representing the layout size. The encoder employs four MSGA modules with scales $s$ progressively increasing as $[1\times, 2\times, 4\times, 8\times]$ to capture a broader range of local features. Sparse convolutions are then applied to extract local features $\tilde{H}_2$, which are remapped back to the original sequence order as $H_2$, based on the indices of $X_v$, formulated as follows:

$$\tilde{H}_1 = \text{ALIGN}(H_1, s), \quad \tilde{H}_1 \in \mathbb{R}^{\tilde{w} \times \tilde{h} \times p \times d} \quad (9a)$$

$$\tilde{H}_2 = \text{Scatter\_conv}(\tilde{H}_1), \quad \tilde{H}_2 \in \mathbb{R}^{\tilde{w} \times \tilde{h} \times p \times d} \quad (9b)$$

$$H_2 = \text{REMAP}(\tilde{H}_2), \quad H_2 \in \mathbb{R}^{n_v \times p \times d} \quad (9c)$$

Finally, in the second part of global attention, the input and hidden feature $H_2$ generate the query $Q$, key $K_2$, value $V_2$, with linear projection. Given matrices $Q, Q \in \mathbb{R}^{n_v \times d}$, the output $O$ of MSGA module can be calculated:

$$A_2 = \text{Softmax}(H_2 Q^T), \quad A_2 \in \mathbb{R}^{n_v \times p}, \quad (10a)$$

$$O = A_2^T V_2, \quad O \in \mathbb{R}^{n_v \times d} \quad (10b)$$

The overall computation complexity is reduced from $\mathcal{O}(n^2 d)$ to $\mathcal{O}(npd)$. The extracted cell features in MSGA are fed into the subsequent module for further extraction of topological relationships.

- *Heterogeneous GNN (HGNN)* aggregates node features $X_v$ and $X_u$ in the topological space, as illustrated in Fig. 5(c). The input graph $G$ includes all nodes $V$ and $U$, with $V$'s features derived from the output of the preceding MSGA module and $U$'s features sourced from either the raw input data or the output of the previous HGNN module. We aggregate topological features in three sub-steps, beginning with the transformation of cell and net outputs into hidden representations via two separate single-layer MLP networks.

$$h_v = \text{MLP}_v(O_v), h_u = \text{MLP}_u(O_u) \ or \ \text{MLP}_u(X_u) \quad (11)$$

where $O_v$ and $O_u$ represent the outputs of the previous MSGA and HGNN modules, respectively. We set the dimension of $h_v$ across the encoder's four modules to $\{32, 64, 128, 256\}$ and set $h_v$'s dimension remains 128. The heterogeneous graph message passing layer

*Table 2.* Pre-training and downstream learning parameters, Pred and Esti are abbreviation of Prediction and Estimation.

| Tasks | Lr | Epoch | Weight Decay | Decoder |
|---|---|---|---|---|
| Pre-training | 4e-3 | 100 | 1e-2 | U-Net + MLP |
| Congestion Pred. | 3e-4 | 50 | 1e-4 | U-Net |
| Wirelength Esti. | 4e-4 | 50 | 0 | MLP |

iteratively aggregates topological information, updating cell and net representations. The final cell and net features are generated through MLPs. For example, in cell-to-cell ($v \rightarrow v$) aggregation within $V$, the core mechanism of graph message propagation is described by the following formula:

$$m_v^{(k)} = \text{AGGREGATE}^{(k)} \left( \left\{ h_i^{(k-1)} : i \in \mathcal{N}(v) \right\} \right), \quad (12a)$$

$$h_v^{(k)} = \sigma \left( W^{(k)} \cdot m_v^{(k)} + b^{(k)} \right), \quad (12b)$$

$$O_v = MLP_v(h_v^{(k)}). \quad (12c)$$

where $h_i^{(k-1)}$ is the hidden feature of the cell or net from the $(k-1)$ layer. AGGREGATE$^{(k)}$ is the function that aggregates neighboring node features $\mathcal{N}(v)$, resulting in the aggregated message $m_v^{(k)}$ at layer $k$. The node features are updated using a weight matrix $W^{(k)}$, bias $b^{(k)}$ and activation function $\sigma$. Cell-to-net ($v \rightarrow u$) message propagation is as the same.

After pre-training, the encoder can serve as a powerful neural extractor, providing comprehensive high-level circuit representations to decoders for various downstream tasks.

**Decoder.** The decoder configurations used for both pre-training and fine-tuning are presented in Tab. 2. During fine-tuning, we explore two decoder configurations in our case study, each designed for a specific predictive objective: $1D$-regression, $2D$-regression.

## 4. Experiments

### 4.1. Experimental Setting

We conduct experiments on CircuitNet (Chai et al., 2022; 2023), a large-scale public dataset of IC designs for real-world industrial applications. The CircuitNet-N28 (28nm) version comprises over 10,241 samples from 6 RTL designs, including 54 synthesized netlists with variations in macros, frequencies, and back-end flow settings. In our experimental setup, the pre-training set contains four designs (RISCY-a, RISCY-b, RISCY-FPU-a, RISCY-FPU-b), totaling over 6,000 samples that only utilize the raw input data

Table 3. A comparative analysis of DeepLayout with existing methodologies in congestion prediction.

| Method | 5 samples | | | | 10 samples | | | | 20 samples | | | |
|---|---|---|---|---|---|---|---|---|---|---|---|---|
| | Pearson | MAE | RMSE | SSIM | Pearson | MAE | RMSE | SSIM | Pearson | MAE | RMSE | SSIM |
| GPDL | 0.2679 | 0.2418 | 0.2476 | 0.2926 | 0.2663 | 0.1061 | 0.1117 | 0.5193 | 0.3174 | 0.0662 | 0.0724 | 0.6601 |
| CircuitGNN | 0.2978 | 0.0166 | 0.0520 | 0.7395 | 0.2155 | **0.0128** | 0.0397 | 0.7681 | 0.2374 | 0.0127 | 0.0392 | 0.7693 |
| CircuitPoint | 0.2783 | 0.1061 | 0.1148 | 0.0943 | 0.3000 | 0.0951 | 0.1056 | 0.0899 | 0.1910 | 0.1059 | 0.1161 | 0.0514 |
| Deeplayout | **0.4270** | **0.0146** | **0.0379** | **0.7718** | **0.4383** | 0.0130 | **0.0360** | **0.7820** | **0.4418** | **0.0121** | **0.0349** | **0.7909** |

Table 4. A comparative analysis of DeepLayout with existing methodologies in post-routing wirelength estimation.

| Method | 5 samples | | | 10 samples | | | 20 samples | | |
|---|---|---|---|---|---|---|---|---|---|
| | Pearson | MAE | RMSE | Pearson | MAE | RMSE | Pearson | MAE | RMSE |
| $Net^2$ | 0.1155 | 0.1441 | 0.1849 | 0.3141 | 0.1424 | 0.1766 | 0.2578 | 0.1441 | 0.1793 |
| CircuitGNN | 0.3683 | 0.1323 | 0.1722 | 0.3691 | 0.1371 | 0.1730 | 0.3691 | 0.1319 | 0.1727 |
| DeepLayout | **0.3704** | **0.1305** | **0.1695** | **0.3806** | **0.1290** | **0.1689** | **0.3961** | **0.1270** | **0.1682** |



| Gpdl | w/o pre-train | DeepLayout | Ground Truth |
|---|---|---|---|

Figure 7. Visualization results of congestion prediction.

to represent a large corpus of unlabeled samples. The fine-tuning and test sets each introduce two additional designs, zero-riscy-a and zero-riscy-b. Specifically, the fine-tuning set comprises a small amount of labeled data—configured as 5, 10, or 20 samples—while the test set contains 100 samples. This test set is used to evaluate quality predictions for these two additional designs under various parameter configurations.

## 4.2. Downstream Tasks

DeepLayout is pre-trained on our proposed layout-oriented masking strategy. Predictions from these tasks facilitate targeted early optimizations during the placement stage, thereby reducing dependency on time-intensive EDA tools throughout the design process. Notably, these tasks focus on assessing circuit quality after routing, presenting significant challenges in practical applications.

**Congestion Prediction.** Routing congestion poses a thorny problem in the design process. Addressing this issue early in the placement stage effectively reduces redundant un-routability and minimizes the need for complex routing iterations, and ML methods are widely used to predict congestion throughout the process.

We comprehensively compare the SOTA baseline methods with diverse methods, including image-based, graph-based, point-based machine-learning methods. Compared methods are: 1) Gpdl (Liu et al., 2021), a fully convolutional model

treating layouts as multi-channel images; 2) CircuitGNN (Yang et al., 2022), which integrates geometric information into graphs via a graph neural network; and 3) CircuitPoint (Zou et al., 2023), which uses sparse convolution to perceive spatial relationships, treating standard cells as a point cloud.

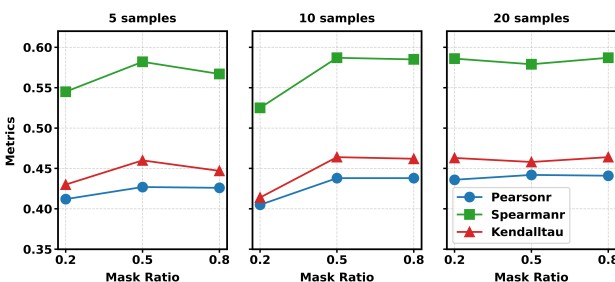

Figure 8. The impact of different mask ratios on congestion prediction.

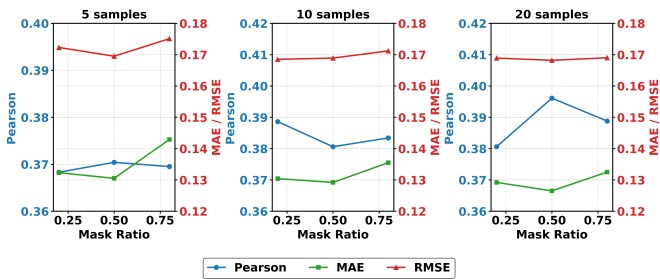

Figure 9. The impact of different mask ratios on post-routing wirelength estimation.

Tab. 3 presents a comparative analysis of DeepLayout and other three prediction methods across different data scales. DeepLayout demonstrates significant performance disparities among the evaluated methods. Fig. 7 shows the examples of congestion prediction results. Deeplayout demonstrates systematic dominance, achieving the highest Pearson

scores (0.4270–0.4418), lowest RMSE (0.0349–0.0379), and best structural fidelity (SSIM: 0.7718–0.7909) across all configurations.

**Post-routing Wirelength Estimation**. Wirelength estimation is a longstanding challenge in the design process, where traditional algorithms often rely on the simpler half-perimeter wirelength (HPWL) metric. However, the more accurate post-routing wirelength, which better models net behavior, remains difficult to estimate effectively.

We comprehensively compare the SOTA baseline methods with diverse methods. Compared methods are: Enhancements to the comparative method $Net^2$(Xie et al., 2020) and improvements to CircuitGNN (Yang et al., 2022) by integrating topological and geometric edges, which enabled more accurate post-routing wirelength estimation.

Tab. 4 evaluates DeepLayout and other post-routing wirelength estimation methods. DeepLayout demonstrates systematic superiority across all metrics, achieving the highest Pearson scores (0.3704–0.3961) and the lowest error rates (MAE: 0.127–0.1305; RMSE: 0.1682–0.1695), with performance improving consistently as dataset size increases.

**Analysis of Mask Ratio.** In this section, we analyze the impact of mask ratio on the performance of downstream tasks. We select three different mask ratios—$20\%, 50\%$, and $80\%$ —to pretrain the DeepLayout framework, followed by fine-tuning on two downstream tasks. As shown in the line graphs in Fig. 8 and Fig. 9, a $50\%$ mask ratio consistently yields the best performance for both routing congestion and wire length estimation, across varying numbers of finetuning samples. This result suggests that circuit layouts exhibit significant differences from images, and therefore, selecting a relatively lower mask ratio leads to more optimal outcomes.

## 5. Conclusion

In this paper, we present DeepLayout, the first circuit representation learning framework tailored for backend design, addressing critical limitations in prior approaches that rely on task-specific models and costly annotated datasets. By integrating a hybrid graph neural network (GNN) and spatial Transformer architecture, our framework effectively captures the topological and geometric characteristics of post-placement circuits while enabling flexible multi-task outputs through a modular decoder design. To overcome labeling bottlenecks, we introduce a self-supervised learning strategy incorporating layout-aware masking mechanisms and dual supervision objectives. Extensive evaluations on industrial-scale benchmarks demonstrate DeepLayout's superiority over specialized state-of-the-art methods in two layout quality assessment tasks, validating its ability to learn intrinsic circuit properties robustly. DeepLayout paves the way for a promising new direction in layout pre-training.

## Acknowledgments

This work was supported in part by the National Key Research and Development Program of China under Grant 2021ZD0114702; in part by the Natural Science Foundation of Beijing, China, under Grant Z230002; in part by the National Science Foundation of China under Grant 62034007; and in part by the 111 Project under Grant B18001; in part by the Hong Kong Research Grants Council (RGC) under Grant No. 14212422 and 14202824.

## Impact Statement

This paper presents work whose goal is to advance the field of Machine Learning. There are many potential societal consequences of our work, none of which we feel must be specifically highlighted here.

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
