# OpenReview forum: "DeepLayout: Learning Neural Representations of Circuit Placement Layout"
_ICML.cc/2025/Conference — ICML 2025 poster_

### Official Review · Reviewer_9pA3 · 2025-02-24

**Overall Recommendation:** 3

**Summary:**

This paper proposes a framework to implement pre-training on circuit netlist and physical layout. Two loss functions are designed to realize the pre-training task. And the pre-training network can be applied to downstream tasks, such as wirelength and congestion prediction.

## update after rebuttal
After carefully reviewing the rebuttals and comments, I would like to maintain my current score.

**Claims And Evidence:**

Yes.

**Essential References Not Discussed:**

No.

**Experimental Designs Or Analyses:**

Yes. There are no issues about the soundness/validity of any experimental designs or analyses.

**Methods And Evaluation Criteria:**

Yes.

**Other Comments Or Suggestions:**

**Typos**
1. $x \in W, y \in H$ below the summation symbol in Eq. 5 is incorrect. Maybe $W$ should be modified to $\text{range}(W)$ or other format?


**Suggestions**
1. Ablation studies of the effectiveness of each pre-training loss and model architecture are not presented. Analyze their seperate effectiveness is encouraged.

**Other Strengths And Weaknesses:**

**Strengths**
1. The presentation of the paper is good, making the approach easy to follow and understand.
2. The idea of masking grid in spatial modality and back to cells in graph modality is novel, which is effectively to capture the spatial and graphic (functional) information.
3. A new supervision signal, the RPA, is proposed to guide the pre-training.
4. Performance on multiple downstream tasks are better than previous baselines, including the congestion and posted-routed wirelength prediction, making the approach a more universal representation learner.


**Weaknesses**
1. The presentation of the pre-training loss should be polished to make it more formal and clearer.
2. Ablation studies about the pre-training loss and the proposed heterogeneous GNN architecture + MSGA are not taken.

**Questions For Authors:**

I have some questions about the calculation of the pre-training loss.
1. For the pre-training task 1 Cell Coordinate Reconstruction, what is the meaning of K in line 266?
2. If there is an one-to-one mapping between the ground-truth coordinate of cell and the predicted coordinate of the cell, why the operation `min' is applied in Eq. 3?
3. For the pre-training task 2 Routing Process Prediction, the definition of $\text{RPA}(x,y)$ in Eq. 4 is not clear. What is the meaning of $x,y$ in the left hand side (LHS), and which $u$ is involved in the calculation of this RPA?
4. In my personal understanding, assume the number of masked grids is $N_m$, the number of nets is $n_u$, and the chip size $W,H$, the RPA for all cases should be a 4D tensor with shape $(N_m \times n_u \times W \times H)$, and $\text{RPA}(i,j,w,h)$ is the RPA: given a specific net $j$, the masked grid $i$ occupies the region $[x_i, y_i] \times [x_i + d_x, y_i + d_y]$, and for each $(w,h) \in [x_i, y_i] \times [x_i + d_x, y_i + d_y]$, the $\text{RPA}(i,j,w,h)$ can be calculated with Eq. 4. If I misunderstanding the calculation, please give a more clearer and formal definition of the RPA.

**Relation To Broader Scientific Literature:**

The pre-trained network could be integrated mature commercial EDA tools to accelerate the design process.

**Theoretical Claims:**

Yes. There are no issues about the correctness of any proofs for theoretical claims.

---

> ### Author Rebuttal · Authors · 2025-04-01
>
> Q1
> 1.Ablation studies of ... is encouraged.
> 2.Ablation studies ... taken.
> A1
> Ablation study of encoder modules
> Congestion prediction
>
> | Method   | 5 | 5 | 5 | 5 | 10 | 10 | 10 | 10 | 20 | 20 | 20 | 20 |
> |---|---|---|---|--|--|--|--|--|---|--|--|--|
> |  | Pearsonr  | MAE | RMSE      | SSIM      | Pearsonr   | MAE        | RMSE       | SSIM       | Pearsonr   | MAE        | RMSE       | SSIM  |
> | train from scratch | 0.1154    | 0.0508    | 0.0874    | 0.1428    | 0.3047     | 0.0289     | 0.0423     | 0.2560     | 0.3398     | 0.0225     | 0.0386  | 0.4513 |
> | wo/ MSGA  | 0.4142    | 0.0288    | 0.0684    | 0.7152    | 0.4271     | 0.0135     | 0.0363     | 0.7755     | 0.4391     | 0.0132     | 0.0356     | 0.7777  |
> | wo/ HGNN   | 0.4138    | 0.0143    | 0.0376    | 0.7670    | 0.4298     | 0.0324     | 0.0728     | 0.7048     | 0.4240     | 0.0335     | 0.0756     | 0.6996 |
> | DeepLayout       | 0.4270    | 0.0146    | 0.0379    | 0.7718    | 0.4383     | 0.0130     | 0.0360     | 0.7820     | 0.4418     | 0.0121     | 0.0349     | 0.7909  |
>
> Post-routing wire length prediction
>
> |Method|5|5|5|10|10|10|20|20|20|
> |------|---------|---------|---------|----------|----------|----------|----------|----------|----------|
> |   |Pearsonr|MAE      |RMSE     |Pearsonr  |MAE       |RMSE      |Pearsonr  |MAE       |RMSE      |
> |train from scratch|0.3342|0.1332|0.1737|0.3048|0.1330|0.1765|0.3233|0.1313|0.1764|
> |wo/MSGA|0.3634|0.1279|0.1712|0.3743|0.1270|0.1706|0.3848|0.1290|0.1684|
> |wo/HGNN|0.3593|0.1311|0.1705|0.3667|0.1326|0.1701|0.3694|0.1303|0.1698|
> |DeepLayout|0.3704|0.1305|0.1695|0.3806|0.1290|0.1689|0.3961|0.1270|0.1682|
>
> Ablation study of pretrain loss:
> Congestion prediction
>
> | Model  | 5| 5|5|10|10|10| 20 |20 |20 |
> |--|-|-|-|-|-|-|-|-|-|
> |  | Pearsonr | MAE  | RMSE | SSIM | Pearsonr | MAE | RMSE | SSIM   | Pearsonr | MAE  | RMSE   | SSIM   |
> |--|-|--|--|--|--|-|--|--|--|--|--|-|
> | train from scratch | 0.1154  | 0.0508 | 0.0874 | 0.1428 | 0.3047   | 0.0289 | 0.0423 | 0.2560 | 0.3398   | 0.0225 | 0.0386 | 0.4513 |
> | task #1 loss| 0.1507  | 0.0454 | 0.0954 | 0.2020 | 0.3203 | 0.0293 | 0.0440 | 0.2534 | 0.3667  | 0.0235 | 0.0387 | 0.3875 |
> | task #2 loss | 0.4050  | 0.0135 | 0.0368 | 0.7760 | 0.3785  | 0.0183 | 0.0455 | 0.7408 | 0.3755 | 0.0141 | 0.0381 | 0.7673 |
> | DeepLayout   | 0.4270  | 0.0146 | 0.0379 | 0.7718 | 0.4383   | 0.0130 | 0.0360 | 0.7820 | 0.4418  | 0.0121 | 0.0349 | 0.7909 |
>
> Post-routing wire length prediction
> | Model | 5|5| 5 | 10|10|10 | 20 |20|20|
> |-|-|-|-|--|-|--|-|-|-|
> |  | Pearsonr | MAE  | RMSE | Pearsonr | MAE | RMSE | Pearsonr | MAE | RMSE |
> | train from scratch | 0.3342 | 0.1332 | 0.1737 | 0.3048 | 0.1330 | 0.1765 | 0.3233 | 0.1313 | 0.1764 |
> | task #1 loss | 0.2737 | 0.1391 | 0.1837 | 0.2878 | 0.1393 | 0.1841 | 0.3064 | 0.1343 | 0.1808 |
> | task #2 loss | 0.3336 | 0.1373 | 0.1761 | 0.3301 | 0.1365 | 0.1779 | 0.3522 | 0.1321 | 0.1737 |
> | DeepLayout  | 0.3704 | 0.1305 | 0.1695 | 0.3806 | 0.1290 | 0.1689 | 0.3961 | 0.1270 | 0.1682 |
>
> Q2
> x∈W,y∈H ... format?
> A2
> We will updating the notation to explicitly use range(W) and range(H).
>
> Q3
> For the pre-training task 1... K in line 266?
> A3
> K denotes the total number of masked cells, where |Iₘ| = K (Algorithm 1, line 6).
>
> Q4
> If there is an one-to-one ... Eq. 3?
> A4
> 'min' indicates the optimization objective to minimize the loss. We will remove this notation from Eq 3 in our paper update.
>
> Q5
> For the pre-training task ... RPA?
> A5
> As shown in Figure 5 (bottom left), RPA is computed for each net u covering area (xul,xuh,yul,yuh) using Eq 4 at every point (x,y) within this rectangle. The final layout RPA is obtained by aggregating values from all nets. We will enhance this explanation with additional visual aids in our final paper to improve clarity.
>
> Q6
> In my personal ... RPA.
> A6
> Our explanation is divided into two parts to address your concerns.
> 1. RPA Feature Computation Without Masking:
> Initially, we compute the RPA feature without considering any masking. Utilizing Eq4 and 5, we perform a global routing probability overlay for all nets across the layout. This process generates a two-dimensional (2D) RPA feature map (w x H), which represents the global routing probability distribution across the entire layout. This feature map encapsulates the inherent routing characteristics of the layout, serving as a foundational element for subsequent processing.
> 2. Self-Supervised Masked Reconstruction:
> Our objective is to enable the network to reconstruct the routing probabilities of masked grid nodes to their original states before masking. To achieve this, we employ a self-supervised reconstruction loss based on Equation 6. Specifically, our loss function calculates the difference between the predicted routing probabilities of the masked grids and their original routing probabilities prior to masking. This approach aligns with the implementation details of Masked Autoencoders (MAE), ensuring consistency and effectiveness in the reconstruction process.

---

### Official Review · Reviewer_PTE9 · 2025-03-09

**Overall Recommendation:** 3

**Summary:**

This paper presents a method to learn the layout representation of circuit for downstream tasks. The method is based on an graph neural network (GNN) and a mask training strategy.

## update after rebuttal
I keep my rating since the authors have answered my question, and I still slightly lean toward accept.

**Claims And Evidence:**

The reported experiment results support the claim of contribution.

**Essential References Not Discussed:**

Not that I know.

**Experimental Designs Or Analyses:**

The experiment seems overall reasonable, and the authors try to perform thorough comparisons.

**Methods And Evaluation Criteria:**

The overall network and loss seems reasonable to me but not novel in my opinion.
It seems borrow some existing method and make it work for circuit design data.

**Other Comments Or Suggestions:**

None.

**Other Strengths And Weaknesses:**

From an outsider (from the circuit design communitiy) point of view:

Strength:
- if the claim of authors is true, the representation learning will be useful for other downstream tasks just like similar thing happen in other field.
- the evaluation results suggest that the proposed method outperform existing works.

Weakness:
- most of the design in the proposed network seems not novel and exists in network/method for other domains. I personally think this is not critical as long as the method is working well for this specific domain..

**Questions For Authors:**

None.

**Relation To Broader Scientific Literature:**

To be honest, since I am not familiar with this community.
So even though I think this work will be useful and impactful for a large circuit design community, I am not sure whether this method will be useful or not..

**Theoretical Claims:**

There is no theoretical claims that requires proofs. Or at least, I believe there is no way to prove the effectiveness or performance of this kind of neural network work.

---

> ### Author Rebuttal · Authors · 2025-04-01
>
> Q1
> “most of the design in the proposed network seems not novel and exists in network/method for other domains. I personally think this is not critical as long as the method is working well for this specific domain..”
>
> A1
>
> We appreciate your thoughtful response and the opportunity to address our concerns more comprehensively. As you have rightly pointed out, we have drawn inspiration from the network architectures and self-supervised learning methods in the field of artificial intelligence in designing DeepLayout. However, our direct adaptation of these methodologies to circuit layout representation learning revealed three critical technical challenges:
>
> 1） Distinct Graph Structures of Circuit Layouts
>
> An integrated circuit's graph structure, derived from Verilog netlists, differs from traditional graph learning domains. It incorporates gate-level physical properties during technology mapping (e.g., TSMC N3), layout positions, and routing interconnect details—unlike point clouds or social networks, as shown in the table below:
>
>
> |                | Social Network Graph | Point Cloud | Circuit Graph Structure |
> |----------------|----------------------|-------------|-------------------------|
> | ​**Node Attributes** | Yes                  | No          | Yes (height, width, power) |
> | ​**Geometric Information** | No               | Yes         | Yes (x, y coordinates)   |
> | ​**Edges**       | Yes                  | No          | Yes (length, resistance, capacitance) |
>
>
> To address this issue, we propose a novel encoder architecture that jointly learns the topological structure and geometric information of the circuit graph structure through the combination of Multi-Scale Graph Attention (MSGA) and Heterogeneous Graph Neural Networks (HGNN). By fully extracting these two types of features, DeepLayout enables comprehensive extraction of topological and spatial features, generating layout representations applicable to routing prediction - a computationally intensive critical phase in circuit design.
>
> 2） Limitations of Self-Supervised Learning in Circuit Layouts
>
> Self-supervised learning in computer vision and graph learning relies on inherent labels for restoration tasks, but this approach has limitations in circuit layout learning. First, circuit graphs have low redundancy, making it hard to restore masked node attributes without extra guidance. Second, layout representation learning aims to predict routing-phase quality, but traditional node-masking methods fail to capture routing characteristics.
>
> |                     | MAE       | GraphMAE2 | DeepLayout                              |
> |---------------------|-----------|-----------|-----------------------------------------|
> | ​**Input**           | Image     | Graph     | Heterogeneous Graph                     |
> | ​**Masking Unit**    | Patch     | Independent Nodes | Set of Nodes in the Grid         |
> | ​**Masking Ratio**   | 75%       | 50%       | 50%                                    |
> | ​**Is there guidance?** | No      | No        | Yes (Topological edges on the graph serve as guidance) |
> | ​**Number of Masks** | Fixed     | Varies with the number of nodes on the graph | Varies with the layout size |
> | ​**Supervisory Signal** | Masked Pixels | Node Attributes | Node Geometric Coordinates + Routing Probability Algorithm (RPA) |
>
>
> To overcome these challenges, we propose a self-supervised learning approach tailored for circuit layouts. Our key design points include: 1) introducing a grid-based masking strategy, 2) masking only the geometric information of nodes while retaining edge information as guidance for restoration, and 3) designing two distinct supervised tasks. Notably, Supervised task #2 utilizes the easily computable Routing Probability Algorithm (RPA) feature as a supervisory signal to captures essential aspects of the routing process.
>
> 3）Downstream Tasks in Circuit Layout Representation Learning
>
> In graph representation learning, downstream tasks typically involve learning global graph attributes or predicting node classifications, without the need for a specialized decoder. However, downstream tasks in circuit layout learning span multiple levels, from individual nodes to 2D representations.
>
> | ​**Hierarchy** | ​**Downstream Tasks in Graph Learning** | ​**Downstream Tasks in Circuit Graph Learning**       |
> |---------------|----------------------------------------|----------------------------------------------------|
> | ​**Node**      | Node Classification                    | Line length of each net after routing              |
> | ​**Edge**      | Whether an edge exists or not          | Timing prediction after routing                    |
> | ​**Whole**     | Graph Classification                   | Overall power consumption prediction              |
> | ​**2D**        | None                                   | Layout congestion prediction, DRC violation prediction |

---

### Official Review · Reviewer_xVSW · 2025-03-13

**Overall Recommendation:** 3

**Summary:**

This paper proposes a representation learning framework for backend circuit design by integrating GNNs and spatial transformers to capture both topological connectivity and geometric distribution of circuits. Also the authors propose a self-supervised learning method based on mask-based autoencoder for layout representation to improve sample efficiency. The results show that the proposed method outperformed SOTA methods in congestion and post-routing wire length estimation.

**Claims And Evidence:**

The proposed Deeplayout provides a general representation learning framework for backend circuit design that captures key circuit attributes without requiring task-specific engineering. The proposed masking strategy preserves key geometric and topological circuit characteristics so that it effectively reconstructs the routing prediction. Tables 2 and 3 provide quantitative comparisons showing Deeplayout outperforming SOTA models in congestion and post-routing wirelength estimation.

**Essential References Not Discussed:**

It could mention other graph-based learning techniques in EDA beyond CircuitGNN and discuss on RL-based methods for backend placement to strengthen the background.

**Experimental Designs Or Analyses:**

The study uses a large-scale public database of IC designs for real-world industrial applications. The authors carried out performance comparison across baselines, ablation study on mask ratio, and provided both statistical and visualization results.

**Methods And Evaluation Criteria:**

The authors evaluated Deeplayout using two downstream tasks, congestion prediction and post-routing wirelength estimation. Also the performance of Deeplayout was compared to four other existing methods.

**Other Comments Or Suggestions:**

Swap Figure 1 and Figure 2 for better readability.

Typo inPage 2, line 93: “Therefor” → “Therefore”.

Text and details in Figure 5 are too small to read clearly.

Visualizations of congestion predictions in Figure 7 are not clear. Please consider improving contrast and resolution.

**Other Strengths And Weaknesses:**

Strengths:
-The paper effectively acknowledges the multi-level dependencies in EDA workflows, considering both upstream and downstream tasks.

-General-purpose representation learning eliminates the reliance on task-specific models, improving flexibility.

-Strong empirical results demonstrate that the proposed method outperforms existing SOTA approaches.

-Well-structured experimental design, with comprehensive benchmarking against multiple baselines.

Weakness:
-Scalability concerns are not addressed.

**Questions For Authors:**

How does DeepLayout perform on larger circuits with millions of components?

What are the computational resource requirements for pretraining DeepLayout?

Can DeepLayout generalize to other layout tasks (e.g., DRC prediction, power estimation)?

Do the authors plan to extend this to full-chip routing prediction?

**Relation To Broader Scientific Literature:**

Application-wise, the paper extends prior work on front-end circuit representation learning to backend design. Methodology-wise, it leverages a masked autoencoder, contrastive learning, and graph-based techniques to enhance layout representation.

**Theoretical Claims:**

The paper modeled the circuits as heterogeneous graphs for general circuit representation adaptable to diverse physical design downstream tasks. The author claims that a masked autoencoder can effectively pre-train circuit layouts, reducing the need for labeled data. The paper also suggests masking strategy and shows that a 50% masking ratio achieves the best trade-off between feature learning and model generalization.

---

> ### Author Rebuttal · Authors · 2025-04-01
>
> Q1
> It could mention other graph-based learning techniques in EDA beyond CircuitGNN and discuss RL-based methods for backend placement to strengthen the background. if we miss any related papers, please feel free to point out. We will incorporate them in the final version.
>
> A1
> We sincerely appreciate your valuable suggestions. In the revised manuscript, we will augment the related work section with a discussion of graph-based learning methods in EDA and reinforcement learning techniques for physical backend design.
>
> Q2
> Swap Figure 1 and Figure 2 for better readability.
> Typo in Page 2, line 93: “Therefor” → “Therefore”.
> Text and details in Figure 5 are too small to read clearly.
> Visualizations of congestion predictions in Figure 7 are not clear. Please consider improving contrast and resolution.
>
>
> A2
> We sincerely appreciate your valuable suggestions. We will incorporate these improvements in the final manuscript, including: 1) making the specified revisions to the content; 2) enhancing the clarity of all visualizations.
>
>
> Q3
> "How does DeepLayout perform on larger circuits with millions of components?"
>
> A3
> We sincerely appreciate your valuable suggestion. However, preparing larger circuit design data is extremely time-consuming. As a result, we have not been able to complete all the experiments related to large circuit designs at present. We will make every effort to update the experimental data related to large designs in the following days and discuss them with you.
>
>
>
>
> Q4
> "What are the computational resource requirements for pretraining DeepLayout?"
>
> A4
> The pre-training phase of DeepLayout is performed on an 8 A800 machine, achieving a per-epoch training time of 16 hours. These implementation specifics will be incorporated into the methodology section of our paper.
>
>
>
> Q5
> "Can DeepLayout generalize to other layout tasks (e.g., DRC prediction, power estimation)?"
>
> A5
>
> Thank you for your valuable suggestions. We have additionally designed post-routing timing prediction task that is directly related to evaluating post-routing layout quality.
>
> Post-Routing Timing Prediction:
> Timing is directly correlated with the chip's performance. The encoder of DeepLayout is initialized with pre-trained weights, while the decoder employs a specially designed graph neural network. We fine-tune the network using regression loss and perform timing prediction after few-shot fine-tuning. Following common practice in related literature, we report widely adopted metrics including Pearsonr, R², and MAE. The results are presented below.
>
> | Model      | 5     |5|5                | 10 |10|10                    | 20|20|20                    |
> |-------|-----|----|-----------|----------------|-------------------------------|------------------|--|--|---------|
> |            | Pearsonr   | R2      | MAE    | Pearsonr   | R2      | MAE     | Pearsonr   | R2      | MAE     |
> | timergcn  | 0.4269     | 0.1434  | 0.1391 | 0.4432     | 0.1647  | 0.1382  | 0.5421     | 0.2598  | 0.1285  |
> | DeepLayout| 0.7644     | 0.5046  | 0.0970 | 0.7974     | 0.5893  | 0.0877  | 0.8101     | 0.5933  | 0.0833  |
>
>
> These experimental results further demonstrate that our proposed pre-training methodology and network architecture can effectively support downstream tasks for characterizing post-routing layout performance.
> Due to the time-consuming nature of data preparation and network training, we were unable to complete all the experiments related to DRC prediction within the limited number of days. We will make every effort to update the experimental data related to large-scale DRC in the following days and discuss it with you.
> Concerning the power prediction task: The current public datasets we employ lack power-related labels and critical information such as input vectors or toggle rates. We therefore designate power prediction as a primary downstream task for our future work.
>
>
> Q6
> "Do the authors plan to extend this to full-chip routing prediction?"
>
> A6
>
> We acknowledge the reviewer’s question regarding whether our prediction method operates on full-chip designs.  Our approach indeed performs post-routing performance prediction on complete circuit chips. Specifically:
> 1.Full-Chip Input Representation: Throughout all stages—pre-training, fine-tuning, and testing—DeepLayout processes  complete circuit graphs that represent the entire chip design.
> 2.Post-routing prediction target: In DeepLayout, the labels for all downstream tasks are extracted after the routing process. In other words, the current objective of DeepLayout is to make routing predictions.
> This distinguishes our work from partial or tile-based prediction approaches, ensuring relevance for real-world chip design optimization.

---

### Official Review · Reviewer_eM95 · 2025-03-14

**Overall Recommendation:** 3

**Summary:**

This paper proposed a mask-based approach for circuit layout representation learning. Specifically, a grid-based partition is dedicated to dealing with the mask operation on layout. Two tasks are utilized to illustrate the potential, including wirelength estimation and congestion prediction. The results outperform previous approaches.

**Claims And Evidence:**

Yes

**Essential References Not Discussed:**

No.

**Experimental Designs Or Analyses:**

Yes. The experiments are conducted on public benchmarks and compared with previous methods. Meaningful data is reported.

**Methods And Evaluation Criteria:**

Yes

**Other Comments Or Suggestions:**

1. Line 92, "Therefore"
2. Algorithm 1, line 13, should the index i be involved?

**Other Strengths And Weaknesses:**

Congestion and wirelength are typical proxies for ultimate metrics of a chip, such as power, performance, and area. Conducting experiments targeting these ultimate metrics may further strength the paper.

**Questions For Authors:**

1. A 50% mask ratio leads to the best performance, and not the higher the better. Any analysis or insights for this?
2. I'm wondering if the performance difference has any potential correlation to the backbone networks used.
3. Robustness. For a specific mask ratio, if multiple experiments are conducted, what's the variation in the performance?
4. Will the code be publicly available for reproduction?

**Relation To Broader Scientific Literature:**

Not applicable.

**Theoretical Claims:**

No theoretical claims are involved.

---

> ### Author Rebuttal · Authors · 2025-04-01
>
> R1
> Q1:
> "Congestion and wirelength ... strength the paper."
>
> A1
> We have additionally designed post-routing timing prediction task that is directly related to evaluating post-routing layout quality.
> Timing is directly correlated with the chip's performance.  The results are presented below.
> | Model     | 5| 5|5     | 10|10|10 | 20    |20    |20    |
> |--|--|----|--|--|---|---|--|-|---|
> |   | Pearsonr   | R2      | MAE    | Pearsonr   | R2      | MAE     | Pearsonr   | R2      | MAE     |
> | timergcn  | 0.4269     | 0.1434  | 0.1391 | 0.4432     | 0.1647  | 0.1382  | 0.5421     | 0.2598  | 0.1285  |
> | DeepLayout| 0.7644     | 0.5046  | 0.0970 | 0.7974     | 0.5893  | 0.0877  | 0.8101     | 0.5933  | 0.0833  |
>
> Area prediction is unnecessary since post-routing layout area remains nearly unchanged from placement, allowing direct calculation.
> The current public datasets we employ lack power-related labels and critical information such as input vectors or toggle rates. We therefore designate power prediction as a primary downstream task for our future work.
>
> Q2:
> Line 92, "Therefore"
> Algorithm 1, line 13, should the index i be involved?
>
> A2：
> We appreciate your identification of this typo in the manuscript. additionally, in  Line 12, "i" should be corrected to "vi," and in  Line 13, "vk" should be updated to "vi." These corrections will be reflected in the final version of the paper.
>
> Q3
> "A 50% mask ratio ... for this?"
>
> A3:
> We hypothesize that this difference arises due to both the nature of circuit data and our specialized masking strategy:
> From the data aspect, Images tend to exhibit a high level of information redundancy, while circuits contain rich information among sub-circuits, which is not easy to restore from the neighbour.
> From the pre-training method aspect, DeepLayout implements mask nodes within selected grid cells, where each grid size is significantly smaller than the image patches (e.g., 16×16 in MAE). Meanwhile, each grid contains multiple graph nodes, different from the individual node mask style of graph SSL.
> Thus, the optimal 50% mask ratio emerges as an intermediate value between these two modalities. Due to character limitations, I had to shorten the content. If you have other questions, we can discuss them below.
>
> Q4
> "I'm wondering ... networks used."
> A4
> Ablation study of encoder modules
> Congestion prediction
>
> | Method   | 5 | 5 | 5 | 5 | 10 | 10 | 10 | 10 | 20 | 20 | 20 | 20 |
> |---|---|---|---|--|--|--|--|--|---|--|--|--|
> |  | Pearsonr  | MAE | RMSE      | SSIM      | Pearsonr   | MAE        | RMSE       | SSIM       | Pearsonr   | MAE        | RMSE       | SSIM  |
> | train from scratch | 0.1154  | 0.0508    | 0.0874 | 0.1428 | 0.3047 | 0.0289 | 0.0423  | 0.2560 | 0.3398 | 0.0225 | 0.0386  | 0.4513 |
> | wo/ MSGA  | 0.4142 | 0.0288   | 0.0684    | 0.7152    | 0.4271 | 0.0135 | 0.0363  | 0.7755 | 0.4391 | 0.0132 | 0.0356  | 0.7777  |
> | wo/ HGNN   | 0.4138    | 0.0143 | 0.0376    | 0.7670    | 0.4298  | 0.0324  | 0.0728  | 0.7048| 0.4240 | 0.0335  | 0.0756 | 0.6996 |
> | DeepLayout  | 0.4270 | 0.0146 | 0.0379   | 0.7718  | 0.4383  | 0.0130  | 0.0360  | 0.7820 | 0.4418  | 0.0121  | 0.0349 | 0.7909  |
>
> Post-routing net length prediction
>
> |Method|5|5|5|10|10|10|20|20|20|
> |------|---------|---------|---------|----|--|--|-|----|---|
> |   |Pearsonr|MAE      |RMSE     |Pearsonr  |MAE       |RMSE      |Pearsonr  |MAE       |RMSE      |
> |train from scratch|0.3342|0.1332|0.1737|0.3048|0.1330|0.1765|0.3233|0.1313|0.1764|
> |wo/MSGA|0.3634|0.1279|0.1712|0.3743|0.1270|0.1706|0.3848|0.1290|0.1684|
> |wo/HGNN|0.3593|0.1311|0.1705|0.3667|0.1326|0.1701|0.3694|0.1303|0.1698|
> |DeepLayout|0.3704|0.1305|0.1695|0.3806|0.1290|0.1689|0.3961|0.1270|0.1682|
>
> Q5
> "Robustness. ... performance?"
> A5
> In response, we adopted a  50% mask ratio and conducted three repeated pre-training runs, followed by evaluations on two downstream tasks.
> Congestion prediction:
> | Samples   | Pearsonr (Mean ± Std)    | MAE_1D (Mean ± Std)     | RMSE_1D (Mean ± Std)    | SSIM (Mean ± Std)       |
> |----|--|---|--|---|
> | 5 samples | 0.4270 ± 0.0002    | 0.0138 ± 0.0000 | 0.0371 ± 0.0001 | 0.7743 ± 0.0005  |
> | 10 samples| 0.4393 ± 0.0015          | 0.0128 ± 0.0002          | 0.0356 ± 0.0002          | 0.7828 ± 0.0020          |
> | 20 samples| 0.4412 ± 0.0011          | 0.0123 ± 0.0001          | 0.0351 ± 0.0001          | 0.7890 ± 0.0013          |
>
> Post-routing wire length prediction
> | Samples    | Pearsonr (Mean ± Std)  | MAE_1D (Mean ± Std)   | RMSE_1D (Mean ± Std)  |
> |--|---|--|---|
> | 5 samples  | 0.3688 ± 0.0039 | 0.1331 ± 0.0009   | 0.1698 ± 0.0004  |
> | 10 samples | 0.3830 ± 0.0027        | 0.1316 ± 0.0020       | 0.1690 ± 0.0002       |
> | 20 samples | 0.3949 ± 0.0016        | 0.1313 ± 0.0012       | 0.1681 ± 0.0004       |
>
> Q6
> Will the code ... reproduction?
> A6
> We promise to open-source the DeepLayout code upon paper acceptance.

---

### Decision · Program_Chairs · 2025-05-01

**Decision:**

Accept (poster)

**Comment:**

This paper proposes DeepLayout, a self-supervised learning framework for circuit layout representation that integrates graph neural networks (GNNs) and a masked autoencoder (MAE) strategy. The method introduces a grid-based masking approach tailored to circuit layouts, combining geometric and topological information through a heterogeneous GNN encoder. It demonstrates strong empirical performance on key downstream tasks—congestion prediction, wirelength estimation, and post-routing timing. By reducing reliance on labeled data and offering a generalizable representation, the work bridges machine learning and electronic design automation (EDA), providing a practical tool for chip design optimization.

Reviewers acknowledged the paper’s strengths but raised concerns about: 1. limited architectural novelty, as the GNN and masking components build on existing techniques; 2. scalability and generalization, with questions about performance on larger circuits (millions of components) and other downstream tasks (e.g., DRC, power estimation); 3. ablation and robustness, as pre-training losses and encoder modules lack ablation studies and the performance variability across masking ratios needs justification. The authors addressed most concerns effectively during the rebuttal phase. Some additional experiments were provided, including post-routing timing prediction, ablation studies, and robustness tests. The masking ratio is justified via circuit-specific redundancy analysis. However, there exist some remaining gaps, such as theoretical insight into masking ratio trade-offs and full-chip routing prediction results.

Overall, this paper presents a practically valuable framework for circuit layout learning, with strong empirical results and thorough rebuttals. After rebuttal, all the reviewers leaned toward acceptance. Thus, I recommend acceptance and highly recommend the authors to revise the paper based on all the reviews.